# Transplantation of discarded livers following viability testing with normothermic machine perfusion

Hynek Mergental [1,2,3,11✉], Richard W. Laing[1,2,3,11], Amanda J. Kirkham[4], M. Thamara P. R. Perera[1], Yuri L. Boteon [1,2,3], Joseph Attard[1,2,3], Darren Barton [5], Stuart Curbishley[2,3], Manpreet Wilkhu[5], Desley A. H. Neil [2,3,6], Stefan G. Hübscher[2,3,6], Paolo Muiesan[1], John R. Isaac[1], Keith J. Roberts[1,3], Manuel Abradelo[1], Andrea Schlegel[1,3], James Ferguson[1,2], Hentie Cilliers[1], Julian Bion[7], David H. Adams[1,2,3], Chris Morris [8], Peter J. Friend[8,9], Christina Yap [4,10,11], Simon C. Afford[2,3,11✉] & Darius F. Mirza [1,2,3,11✉]

There is a limited access to liver transplantation, however, many organs are discarded based on subjective assessment only. Here we report the VITTAL clinical trial (ClinicalTrials.gov number NCT02740608) outcomes, using normothermic machine perfusion (NMP) to objectively assess livers discarded by all UK centres meeting specific high-risk criteria. Thirty-one livers were enroled and assessed by viability criteria based on the lactate clearance to levels ≤2.5 mmol/L within 4 h. The viability was achieved by 22 (71%) organs, that were transplanted after a median preservation time of 18 h, with 100% 90-day survival. During the median follow up of 542 days, 4 (18%) patients developed biliary strictures requiring re-transplantation. This trial demonstrates that viability testing with NMP is feasible and in this study enabled successful transplantation of 71% of discarded livers, with 100% 90-day patient and graft survival; it does not seem to prevent non-anastomotic biliary strictures in livers donated after circulatory death with prolonged warm ischaemia.

[1] Liver Unit, Queen Elizabeth Hospital, University Hospitals Birmingham NHS Foundation Trust (UHBFT), Birmingham, UK. [2] National Institute for Health Research (NIHR), Birmingham Biomedical Research Centre, University of Birmingham and University Hospitals Birmingham NHS Foundation Trust, Birmingham, UK. [3] Centre for Liver and Gastrointestinal Research, Institute of Immunology and Immunotherapy, University of Birmingham, Birmingham, UK. [4] Cancer Research UK Clinical Trials Unit, University of Birmingham, Birmingham, UK. [5] D3B team, Cancer Research UK Clinical Trials Unit, University of Birmingham, Birmingham, UK. [6] Department of Cellular Pathology, Queen Elizabeth Hospital, University Hospitals Birmingham NHS Foundation Trust (UHBFT), Birmingham, UK. [7] Department of Intensive Care Medicine, University of Birmingham, Birmingham, UK. [8] OrganOx Limited, Oxford, UK. [9] Nuffield Department of Surgical Sciences, University of Oxford, Oxford, UK. [10] Clinical Trials and Statistics Unit, The Institute of Cancer Research, London, UK. [11] These authors contributed equally: Hynek Mergental, Richard W. Laing, Christina Yap, Simon C. Afford, Darius F. Mirza. ✉email: hynek.mergental@uhb.nhs.uk; S.C.AFFORD@bham.ac.uk; darius.mirza@uhb.nhs.uk

L iver transplantation is a life saving treatment for selected patients with end-stage liver disease, primary liver cancer and fulminant hepatic failure. The incidence of liver disease has risen by 500% over the last 4 decades, however, access to transplantation is limited by the shortage of donor organs[1]. As a consequence, 240 patients (19%) waiting for liver transplantation in the United Kingdom either died or were removed from the waiting list in 2016–2017[2]. Data from the United States shows a similar pattern, comprising 32% of those listed for transplant (3629 patients) within 3 years of listing[2,3]. The demand for liver grafts has driven the wider use of extended criteria donors[4]. However, these are associated with an increased risk of primary non-function or delayed failure[5–9], and the acceptance of these higher-risk organs varies widely[10]. Because of these inferior outcomes, and the difficulty of predicting organ viability, many potential donor organs remain unutilised. The high waiting list mortality justifies the utilisation of more marginal grafts, but current practice requires risk mitigation by matching high-risk livers to lower-risk recipients to achieve patient survival rates that are acceptable[11]. Furthermore, the determination of suitability of a graft for transplantation largely depends on a surgeon's subjective assessment of the graft's appearance, using criteria that are known to be unreliable[12].

Organ preservation currently relies upon cooling to ice temperature to reduce cellular metabolism, and infusing specialist solutions to limit cellular damage. Oxygen deprivation and accumulation of by-products of anaerobic metabolism limit the duration of storage and result in ischaemia–reperfusion injury at the time of implantation. This process is more severe in marginal organs[13]. Normothermic machine perfusion (NMP) has been shown to reduce preservation-related graft injury compared to static cold storage in transplantable livers, according to current selection criteria, in a prospective European trial, which also demonstrated increased utilisation of organs[14]. In NMP, the liver is supplied with oxygen, nutrients and medication at physiological temperature and pressures, maintaining conditions that support homoeostasis, normal metabolic activity and objective assessment of function in real-time. Experimental data have shown that end-ischaemic NMP facilitates replenishment of adenosine triphosphate and glycogen levels. Based on increasing clinical experience, viability criteria have emerged; these are objective parameters, measurable during NMP[15]. Whilst the feasibility of this approach has been demonstrated in a proof-of-concept series, it has not been validated in a rigorous clinical trial[16,17].

We therefore conducted this prospective, non-randomised, adaptive phase 2 trial in a large single centre, to evaluate the potential of NMP to provide objective assessment of the viability of livers currently deemed unsuitable for transplantation, and to transplant those that met predetermined criteria. The primary clinical objective underlying this project was the increased and safe utilisation of livers which are currently discarded.

The trial demonstrates that viability testing with NMP is feasible, and the objective assessment enables successful transplantation of 71% of perfused discarded livers, with 100% 90-day patient and graft survival. The intervention does not seem to prevent the development of non-anastomotic biliary strictures in livers donated after circulatory death (DCD) with prolonged donor warm ischaemic times.

One hundred and sixty-four patients on the waiting list were approached for potential participation, of which 53 were consented, and 22 were enroled in the study and received rescued grafts. The potential participants were counselled regarding the high-risk nature of the project and unknown long-term outcomes of resuscitated livers. As a consequence, a proportion of patients were understandably reluctant to participate, and therefore the lack of suitable consented recipients was the principal rate limiting factor for inclusion. The number of consented patients at any given time ranged from 1 to 9; the flow diagram displaying the progress of patients through the trial is shown in Fig. 2.

**Donor liver characteristics and liver biopsy features**. In 8 (26%) donors the liver was the only procured organ. All discarded donor livers entered in the study satisfied one or more of the inclusion high-risk criteria. The livers enroled in the trial consisted of 17 organs donated after brainstem death (DBD) and 14 DCD. Many of these organs looked grossly suboptimal, with some degree of steatosis, capsular fibrosis or rounded edges with multifactorial reasons for discard, that was captured by the donor risk index (DRI) > 2.0 in 22 (71%) livers, with the median DRI 2.2 (1.9–2.9). Detailed characteristics are shown in Table 1 and Supplementary Table 1. Photos of all included livers are presented in Fig. 3. The transplanted livers were typically smaller than non-viable ones (1.7 vs. 2.0 kg, $p = 0.015$; Kruskal–Wallis test), with lower peak pre-mortem donor liver enzyme levels. The median static cold storage time before starting NMP was 7 h:44 min (6:29–10:25). Only 3 (10%) livers were included in the trial primarily for macrosteatosis >30%, (50%, 80% and 60% macrovesicular steatosis combined with 11 h:55 min, 12 h:00 min and 6 h:15 min cold ischaemia, respectively). Glycogen content and steatosis degree did not predict the viability assessment results. The detailed histological finding of each study liver is provided in Supplementary Table 2.

**Perfusion parameters assessment**. During the NMP procedure 25 livers quickly recovered metabolic activity and cleared lactate to the target level (details provided in Fig. 4). A biopsy of a suspicious donor colonic lesion confirmed malignancy, making one liver unsuitable for transplantation, after meeting the viability criteria. In three livers, criteria were initially met, however, metabolic function thereafter deteriorated within the first 4 h, with increasing lactate. In two cases the transplant procedure was not commenced and the livers were discarded. In the third, the explant had begun, and the procedure continued. Overall, 22 (71%) livers met the viability criteria and were transplanted following a median total preservation time of 17 h:53 min (16:17–21:48; Table 2).

**The study patients**. The majority (64%) of recipients were men, and median age was 56 (46–65) years. The leading indication for transplantation was alcohol-related liver disease (36%), followed by primary sclerosing cholangitis (27%) and non-alcoholic steatohepatitis (18%). In three (14%) patients the underlying liver disease was complicated by liver cancer. The median UKELD[18] score was 52 (49–55), with a calculated laboratory MELD score of 12 (9–16). Details are provided in Table 2 and Supplementary Table 3.

**Co-primary study outcomes**. Thirty-one livers were enroled into the trial for objective assessment by NMP. Twenty-two of these livers met the viability criteria and were transplanted, resulting in a significant successful rescue rate of 71% (22/31, 90% Wilson CI: 56.3–82.2%), to conclude that the procedure is feasible. All 22

## Results
### Characteristics of discarded liver offers and study participants.
Over the 16-month study duration from November 2016 to February 2018, there were 185 livers discarded for clinical use and offered for research. Characteristics of those offers and the study inclusion flowchart are provided in Fig. 1a, b.

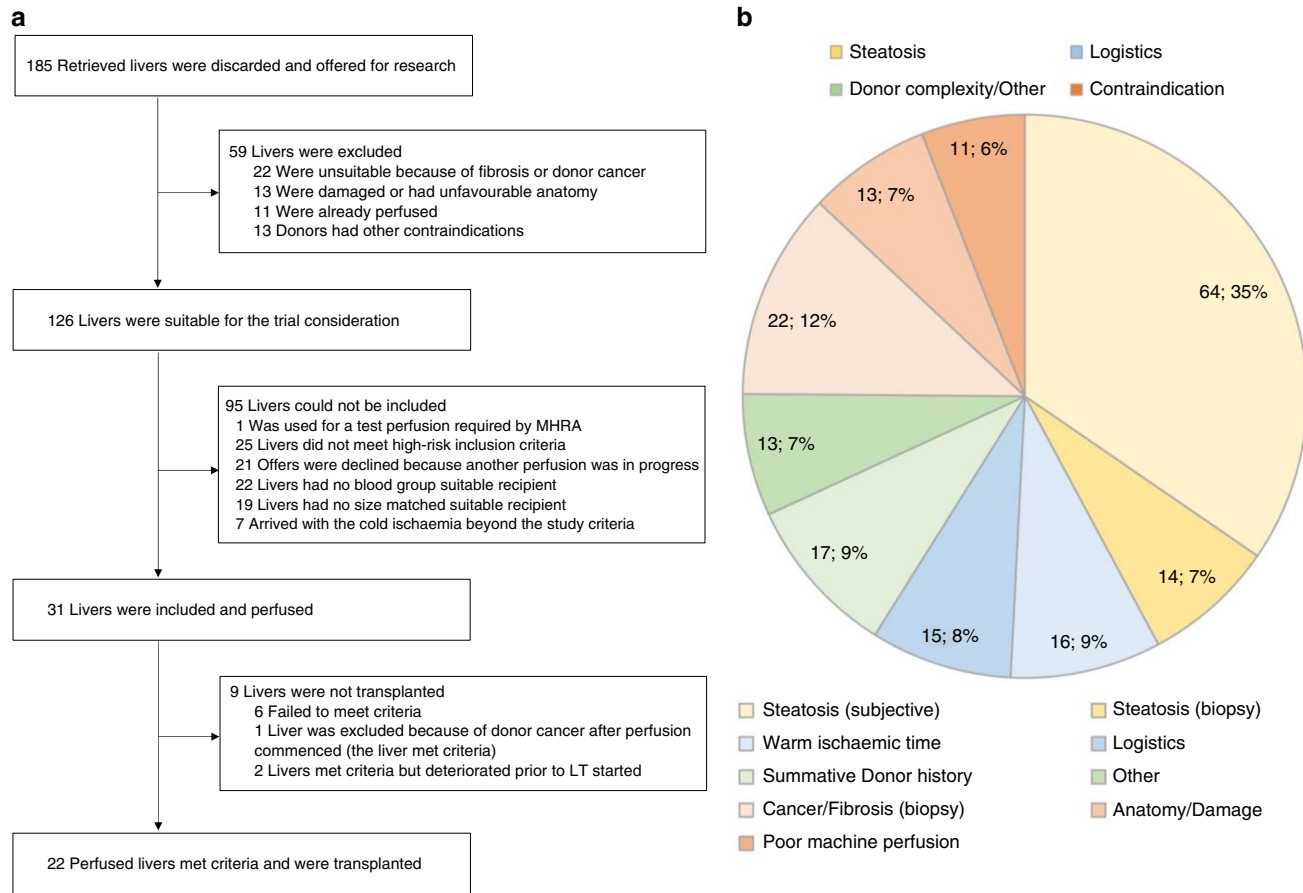

**Fig. 1 Information about discarded livers in the UK between November 2016 and February 2018. a** The study livers inclusion flowchart. Over the 16-month study period there were 185 discarded liver research offers, of which 59 (32%) were not eligible for the trial due to an incidental finding of cancer, macroscopically apparent cirrhosis or advanced fibrosis, severe organ damage or previous machine perfusion. There were 126 livers suitable for the trial, with steatosis being the leading cause of organ discard with 78 (42%) offers. Stringent donor inclusion criteria were not met in 25 (14%) offers and on 21 (11%) occasions the research team was already committed to the perfusion of another study liver. A liver was considered for the trial only if it could be allocated to a consented, potential blood group- and size-matched low-risk recipient. Many recipients were apprehensive to participate in such a high-risk clinical trial, and as a consequence, at any given time there were usually only one to three patients consented. A significant proportion of approached patients declined to take part, or were transplanted with a standard quality liver before agreeing to take part in this study. Eventually, thirty-one livers were enroled to the trial, of which 22 (71%) grafts met the viability criteria and were successfully transplanted. **b** A summary of reasons for livers being discarded in the United Kingdom between November 2016 and February 2018. A total of 64 livers were discarded for severe steatosis on visual assessment, with 14 discarded for severe steatosis based on urgent liver biopsy. A percentage of livers were declined due to intra-abdominal or lung malignancies (e.g. colonic cancer in donor 22). This did not include primary brain tumours or small renal cell cancers which are almost always considered for donation. The reasons for logistic discard, include the transplant team already being committed to one or more transplantations, lack of a suitable recipient or too long an anticipated cold ischaemia time due to delays with transportation.

(100%) transplanted patients were alive at day 90 post-transplantation—greater than the 18/22 required by the trial design.

**Transplant outcomes**. Graft 90-day survival was 100%. Seven (32%) patients developed early allograft dysfunction, and 7 (32%) patients developed Clavien–Dindo complication grade ≥3, including 4 (18%) cases with acute kidney injury requiring renal replacement therapy. The median intensive care and in-hospital stays were 3.5 days (3–4) and 10 days (8–17), respectively. The 1-year patient and graft survival were 100% and 86%, respectively. Details are provided in Table 3.

**Vascular and biliary complications**. One patient developed an intra-operative hepatic artery thrombosis after receiving a DBD graft that had sustained a hepatic arterial injury during procurement. The artery was reconstructed but post-operatively thrombosed, undergoing emergency revascularisation which achieved long-lasting arterial patency. The graft, however, developed biliary strictures requiring multiple interventions and eventual re-transplantation.

The per-protocol magnetic resonance cholangiopancreatography (MRCP) imaging at 6 months revealed that 2 (9%) patients developed anastomotic, and 4 (18%) patients non-anastomotic biliary strictures that presented with cholestatic symptoms. With the exception of the patient with hepatic artery thrombosis, all biliary strictures affected recipients of DCD grafts. During the study median follow up of 542 days (456–641), 4 patients underwent liver re-transplantation (at day 120, 225, 375 and 417). The details are provided in Table 3 and Supplementary Table 3.

**Comparison of outcomes with contemporary matched controls**. Patient and graft survival rates at 12 months (100% and 86%, respectively) were similar to the matched controls (96% and 86%, respectively) as shown in Fig. 5. The incidence of early allograft dysfunction was higher in the study group (32% vs. 9%, odds ratio 5.6, 95% CI: 1.1–27.8, $p = 0.034$; conditional logistic

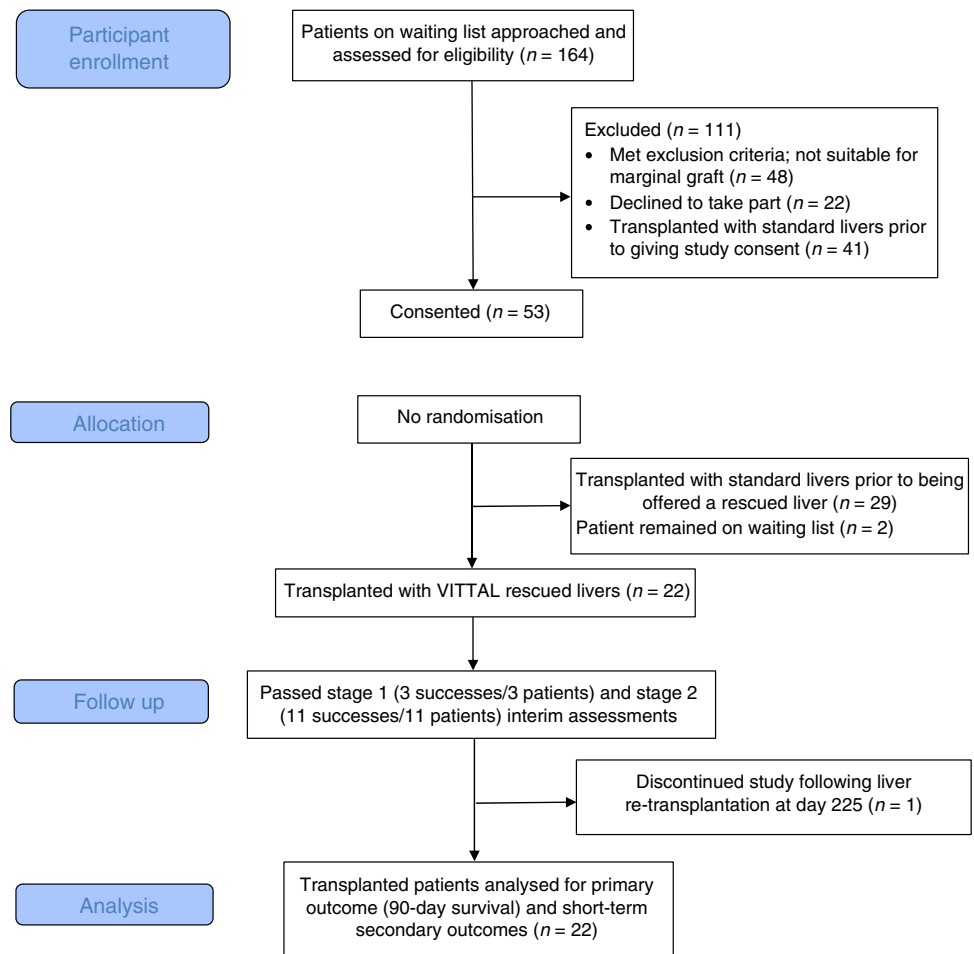

**Fig. 2 CONSORT flow diagram displaying the progress of patients through the trial.** One hundred and sixty-four patients on the waiting list were approached for potential trial participation. Of those, 111 were excluded; 48 patients met exclusion criteria and were not suitable for a marginal liver graft. Twenty-two patients declined to take part and 41 patients either received a transplant before they provided study consent, or were de-listed, or subsequently met exclusion criteria. Eventually 53 patients consented to the study, of which 29 underwent transplantation with a standard quality liver allocated outside the trial. Twenty-two patients were enroled in the trial and received a salvaged liver.

regression). There were no differences in the other assessed parameters, including the need for post-transplant renal replacement therapy, hospital stay or incidence of Clavien–Dindo grade ≥ 3 complication rates. The incidence of clinically manifest non-anastomotic biliary strictures was higher in the study group (18% vs. 2%, odds ratio 8.0, 95% CI: 0.9–71.6; $p = 0.063$; conditional logistic regression), although this result needs to be interpreted with caution as the matched control patients did not receive systematic bile duct imaging. Due to the small sample sizes these comparison results should be interpreted with caution, and the controls were included to present the study results within the context of the unit's contemporary outcomes. The details are shown in Table 3.

## Discussion

Utilisation of livers from organ donors is currently a major challenge in liver transplantation[19]. Despite a waiting list mortality in Western countries reaching 20–30%, an increasing proportion of extended criteria livers are unused due to concerns of primary non-function and early graft dysfunction[20,21]. The decision to discard donor livers is still largely based upon donor history and subjective assessment by the transplanting surgeon. Standard cold static preservation does not allow for any assessment of liver function, and the only other source of information is liver histology, which is able to diagnose severe large droplet fatty

change, a well-recognised risk factor for non-function[21]. This study has demonstrated that moving from subjective evaluation to objective testing during NMP might salvage a high proportion of those livers that are currently discarded. The need to improve the method by which high-risk livers are assessed was illustrated in this study by the absence of significant differences in the donor characteristics between transplanted and discarded livers.

The present trial is the first to systematically investigate objective viability criteria in livers that met specific high-risk features in organs initially considered "untransplantable"[11,22]. One major challenge addressed in the VITTAL trial design was that each discarded liver had to also fulfil one or more pre-defined objective high-risk criteria, as the considerations for liver transplantability are always multi-factorial, including the recipient's condition, logistical aspects and the surgeon's (or transplant centre's) experience and risk-taking attitude. The utilisation of marginal livers in the United Kingdom was facilitated by the centre-based liver allocation system, allowing the use of high-risk organs in any patient on the waiting list. All enroled organs were simultaneously fast-track offered to all UK transplant centres following the initial decline, and the fact that none of the seven centres were comfortable using any of the livers included in this trial confirms that these organs were uniformly perceived to be of very poor quality. Our team genuinely aimed to push the boundaries of utilisation of the highest risk organs by accessing

**Table 1 Donor and liver characteristics (median, interquartile range).**

| Donor characteristics | Non-transplanted (n = 9) | Transplanted (n = 22) | Overall (n = 31) | p Value[a] |
|---|---|---|---|---|
| Age in years (range) | 57 (52–60) | 56 (45–65) | 57 (45–63) | 0.948 |
| Sex—n (%) | | | | 0.696 |
| Female | 3 (33.3) | 10 (45.5) | 13 (41.9) | |
| Male | 6 (66.7) | 12 (54.5) | 18 (58.1) | |
| Height (cm) | 174 (172–186) | 170 (165–175) | 170 (166–175) | 0.038 |
| Bodyweight (kg) | 79 (75–88) | 81 (70–90) | 80 (70–90) | 0.662 |
| Body mass index (kg/m$^2$) | 28.7 (24.8–29.1) | 29.3 (26.5–32.4) | 28.7 (24.8–32.1) | 0.372 |
| Liver weight (kg) | 2.0 (1.8–2.4) | 1.7 (1.3–1.9) | 1.8 (1.4–2.0) | 0.015 |
| Peak alanine transferase (IU/ml) | 323 (92–1143) | 48 (33–159) | 83 (36–287) | 0.034 |
| Peak gamma-glutyl transferase (IU/ml) | 169 (107–335) | 80 (42–111) | 92 (57–203) | 0.012 |
| Peak bilirubin (μmol/L) | 10 (10–18) | 11 (7–22) | 11 (8–22) | 0.768 |
| History of excessive alcohol use—n (%) | 5 (55.6) | 5 (22.7) | 10 (32.3) | 0.105 |
| Diabetes mellitus—n (%) | 0 (0.0) | 2 (9.1) | 2 (6.5) | 1.000 |
| Donor type—n (%) | | | | 1.000 |
| Donor after brain death | 5 (55.6) | 12 (54.5) | 17 (54.8) | |
| Donor after circulatory death | 4 (44.4) | 10 (45.5) | 14 (45.2) | |
| Donor warm ischaemic time (min)[b] | 20.0 (15.5–22.5)[b] n = 4 | 22.5 (19.0–35.0)[b] n = 10 | 21.0 (19.0–25.0)[b] n = 14 | 0.394 |
| Quality of in situ flush—n (%) | | | | 0.016 |
| Poor | 3 (33.3) | 4 (18.2) | 7 (22.6) | |
| Fair | 4 (44.4) | 1 (4.5) | 5 (16.1) | |
| Good | 2 (22.2) | 17 (77.3) | 19 (61.3) | |
| Cold ischaemic time (min) | 550 (436–715) | 452 (389–600) | 464 (389–625) | 0.277 |
| Donor risk index[c] | 2.3 (2.0–2.7) | 2.1 (1.9–3.0) | 2.2 (1.9–2.9) | 0.728 |
| Histological steatosis assessment—n (%)[d] | | | | 0.113 |
| <30% steatosis | 2 (22.2) | 13 (59.1) | 15 (48.4) | |
| >30% steatosis | 7 (77.8) | 9 (40.9) | 16 (51.6) | |
| Inclusion criteria[e] | | | | |
| Donor risk index >2.0 | 6 (66.7) | 16 (72.7) | 22 (71.0) | 1.000 |
| Steatosis principal reason to discard[f] | 1 (11.1) | 2 (9.1) | 3 (9.7) | 1.000 |
| High liver transaminases | 3 (33.3) | 2 (9.1) | 5 (16.1) | 0.131 |
| Balanced risk score >9 | Not applicable | 2 (9.1) | Not applicable | Not applicable |
| Extensive cold ischaemic time | 2 (22.2) | 3 (13.6) | 5 (16.1) | 0.613 |
| Extensive donor warm ischaemic time | 0 (0.0) | 3 (13.6) | 3 (9.7) | 0.537 |
| Poor in situ flush | 3 (33.3) | 4 (18.2) | 7 (22.6) | 0.384 |
| Perfusion criteria | | | | |
| Lactate clearance <2.5 mmol/L | 3 (33.3) | 22 (100.0) | 25 (80.6) | <0.0001 |
| pH ≥ 7.30 | 3 (33.3) | 19 (86.4) | 22 (71.0) | 0.007 |
| Presence of bile production—n (%) | 6 (66.7) | 18 (81.8) | 24 (77.4) | 0.384 |
| Bile volume (mL) | 10 (2–18) | 60 (15–99) | 46 (2–90) | 0.100 |
| Glucose metabolism | 4 (44.4) | 20 (90.9) | 24 (77.4) | 0.012 |
| Vascular flows criteria met | 9 (100) | 22 (100) | 31 (100) | Not applicable |
| Homogenous liver perfusion | 7 (77.8) | 22 (100.0) | 29 (93.5) | 0.077 |

*Note*: Body mass index is the weight in kilograms divided by the square of the height in metres.
[a]Groups compared by Kruskal–Wallis test to assess differences in continuous variables and Fisher's exact test for categorical variables. Due to the small sample sizes and that the statistical comparison tests were not powered, these results should be interpreted with caution.
[b]Donor warm ischaemic time is defined as the period from the systolic blood pressure decrease below 50 mmHg to commencing the aortic cold flush; this variable applicable only for donors after circulatory death.
[c]Donor risk index as described by Feng et al.[20].
[d]The steatosis includes large and medium droplets steatosis assessment obtained from post-transplant paraffin sections (this result was not known at the time of the liver inclusion).
[e]Each trial liver had to meet one or more of the following inclusion criteria: donor risk index greater than 2.0; biopsy proven liver steatosis greater than 30%; donor transaminases (aspartate transaminase or alanine transaminase) greater than 1000 IU/mL; warm ischaemic time greater than 30 min in donors after circulatory death; extensive cold ischaemic time (defined as the period between the aortic cold flush to the liver implantation, or commencing the normothermic perfusion) greater than 12 and 8 h for donors after brainstem death and circulatory death, respectively; suboptimal liver flush documented by photograph and a transplant surgeon assessment; balanced risk score greater than 9.
[f]This steatosis variable refers to the study inclusion criteria and the results were known before the transplant based on frozen sections histology assessment.

the benefit of rigorous peer-review and continual oversight within the framework of a clinical trial. We included only organs that our team did not feel comfortable to use otherwise, and this attitude was reflected by the 2-tier liver inclusion process embedded in the trial design, and by the fact that 25 livers, that would very likely meet the transplantability criteria, were not considered for study inclusion. Some of the study livers might have been transplantable if the cold ischaemia was very short and a suitable recipient was waiting, but currently the majority of these organs are discarded. With the introduction of the National Allocation system, logistical constraints exacerbated by static cold storage are increasingly common and prevent the utilisation of a rising proportion of marginal livers. In these circumstances, NMP mitigates the reperfusion process, allowing assessment of the organ during perfusion without exposing patients to the risk of primary non-function. In addition, livers discarded due to haemodynamic instability (during procurement or during the process of brain stem death itself), high liver transaminases or poor in situ flush, benefited from perfusion in a controlled, near physiological environment thereby facilitating their recovery. The potential to recondition the liver in the interval between retrieval and implantation has hitherto not been possible.

An intervention which increases successful utilisation of high-risk livers will transform access to transplantation to meet predicted increasing demand, particularly given trends in donor demographics and declining organ quality[4]. Whilst organ

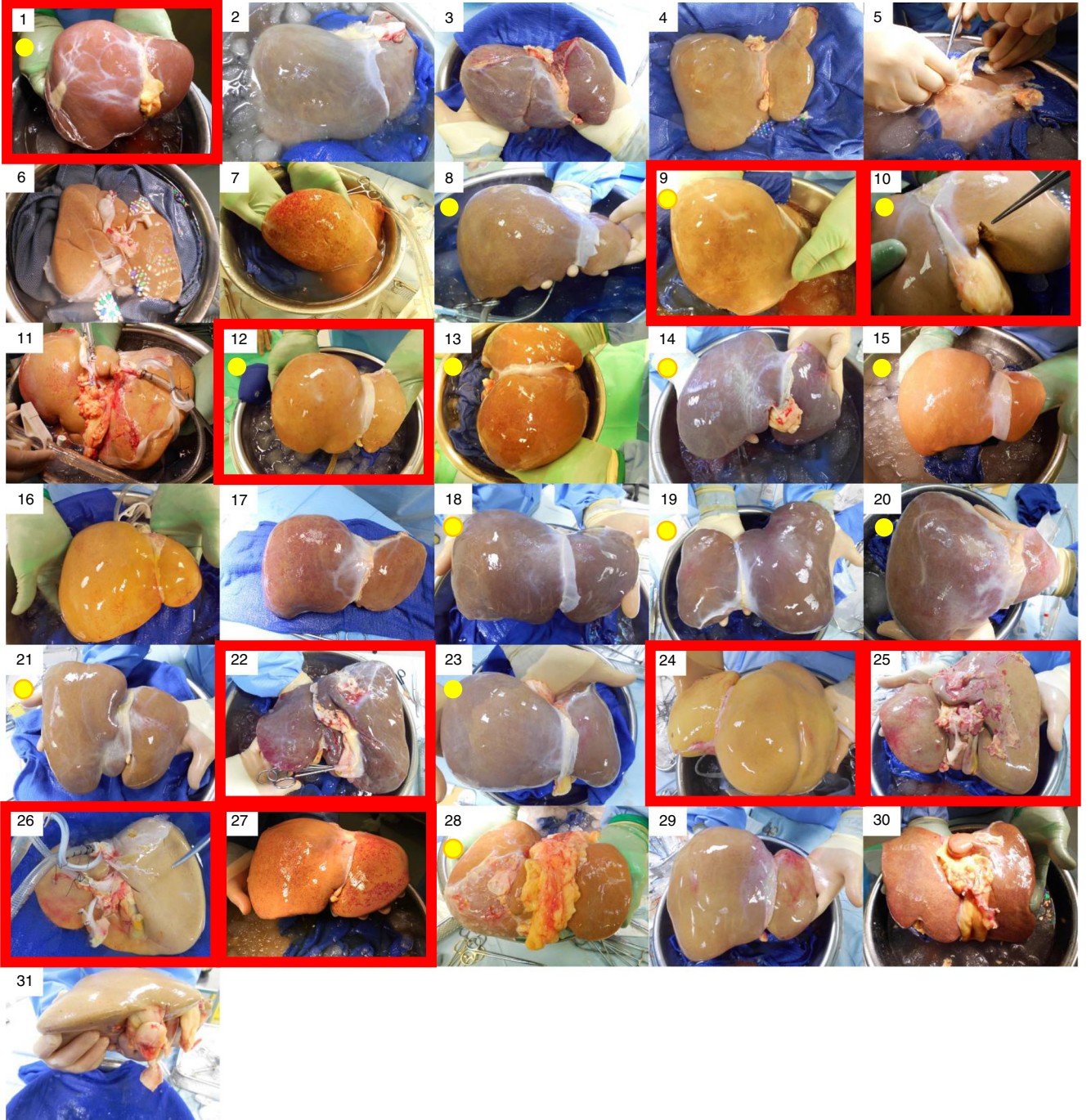

**Fig. 3 The study liver photographs.** The figure shows all 31 livers included in the trial. The red frames designate non-transplanted organs and the yellow dot livers donated after circulatory death.

donation in the UK has increased from 676 to 1149 donors per annum between 2008 and 2018, the proportion of retrieved livers that were discarded has nearly doubled (from 8 to 15%; data from the UK Organ Donation and Transplantation Registry, www.odt. nhs.uk), indicating reluctance of surgeons to accept these organs for their increasingly sicker recipients. In 2017–2018, not only were 174 retrieved livers discarded, but 425 livers from solid organ donors were not even considered suitable for retrieval (11% of DBD and 52% of DCD); it is reasonable to assume that many of these would be suitable for testing with NMP. Salvaging a proportion of these retrieved but discarded organs would add a

good number of transplantable livers annually in the UK, significantly reducing waiting list mortality.

International comparisons demonstrate regional variations in donor demographics and there is evidence that in countries with higher initial organ acceptance rates there is also a higher discard rate, particularly for older donors[23,24]. Viability testing provides objective evidence of liver function with clearance of metabolic acidosis, vascular flows, glucose parameters and bile production; these give the transplant surgeon the confidence to use these organs safely, and minimises the physical and emotional impact of non-transplantation for patients.

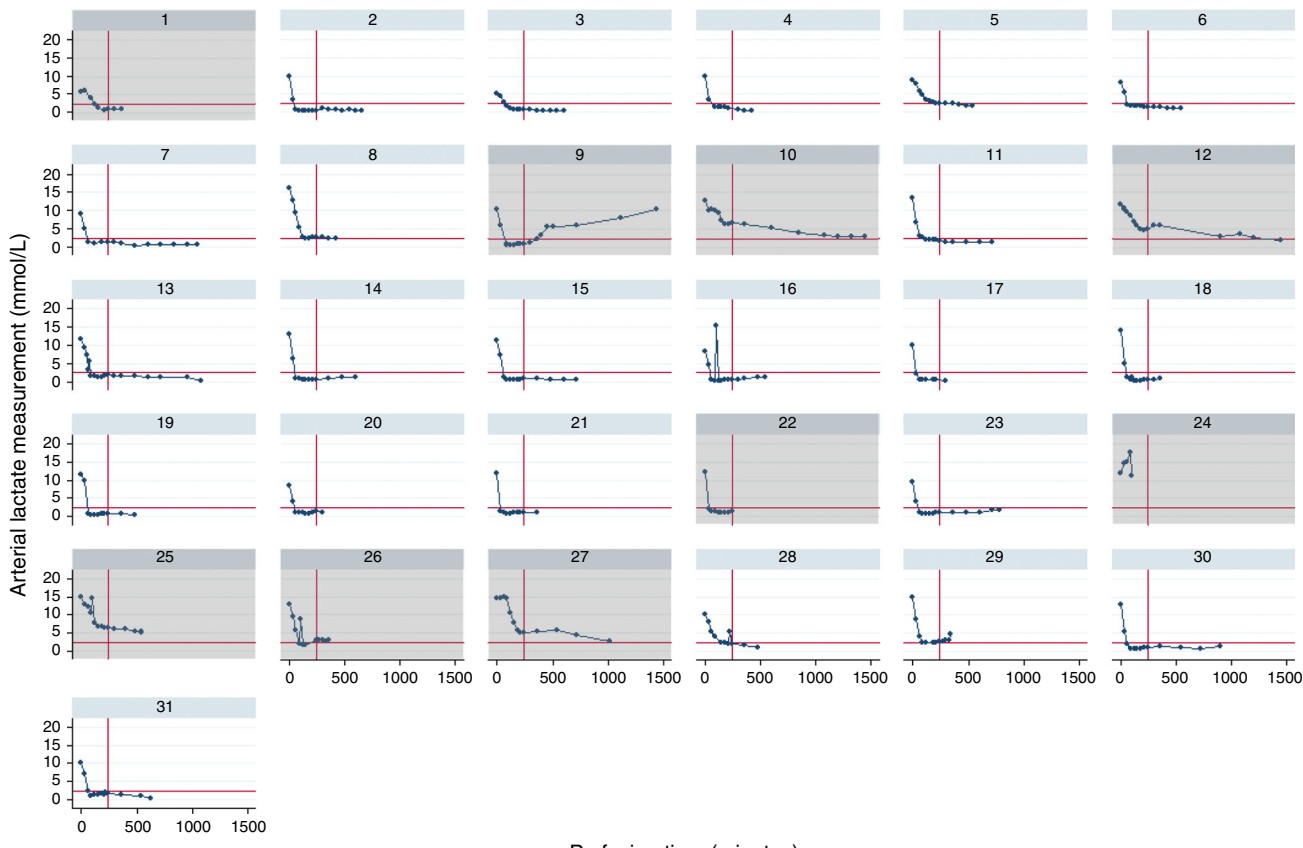

**Fig. 4 The study liver lactate clearance.** Plots of individual liver arterial lactate clearance measured during the NMP perfusion, showing transplantation eligibility thresholds with red lines for lactate levels less than or equal to 2.5 mmol/L. Graphs with grey shading designate livers that were not transplanted. Liver number 22 was from a donor that was unexpectedly diagnosed with a cancer following organ donation.

In the presented study the NMP was commenced following a median cold storage time reaching 8 h. Whilst this approach may simplify adoption of the NMP technology without compromising outcomes in transplantable livers[25], recovery of organs from donors with multiple high-risk features might be further facilitated by limiting cold ischaemia through commencing the perfusion immediately after procurement in the donor hospital[14]. Inevitably there will always be livers that are not suitable for transplantation, demonstrated by 30% of offers with macroscopic cirrhosis, biopsy-proven fibrosis or an incidental finding of donor cancer. A similar proportion of the livers, however, did not meet any of our high-risk criteria and were therefore considered "too good" for inclusion. It is reasonable to assume that NMP assessment would have provided the reassurance needed to justify transplantation in this group as well.

Improvements in transplant logistics is one of the major advantages of NMP[14,25,26], and the study allowed for the machine perfusion duration to be between 4 h (time needed for the viability assessment) and 24 h (maximum recommended time by the perfusion device manufacturer). Once the liver met the viability criteria we aspired to commence the transplantation as soon as possible; however, the perfusion was often extended to allow for a day-time procedure, or to facilitate transplant logistics in the unit. From our experience, 4–6 h perfusion seems to be sufficient for adequate assessment and replenishment of the organ's energy resources. Due to recirculation of metabolites accumulated in the organs during cold ischaemia, the high-risk organs probably do not benefit from prolonged perfusion. The impact of NMP duration on livers initially exposed to prolonged cold ischaemia is an area of our ongoing research interest.

Transplant surgeons in many countries are expanding the donor pool with the use of organs donated after circulatory death[27]. In the context of liver transplantation, the longevity of these organs might be compromised by development of non-anastomotic biliary strictures[8]. The incidence of clinically manifest non-anastomotic biliary strictures in the DCD grafts cohort was 30% (3 out of 10 grafts), higher than the study matched controls group, but similar to other reported high-risk DCD series[28]. In concordance with the European prospective normothermic preservation trial, our results suggested that MRCP findings are likely to over-estimate the incidence of biliary complications[14]. The per-protocol investigation at the 6-month time point would identify over 80% of the clinically relevant biliary strictures and asymptomatic irregularities with varying clinical significance[28]. The presented findings are accurate, as the images were correlated with clinical reviews and liver function tests through the median follow up of 542 (range: 390–784) days. Nevertheless, it is clear that end-ischaemic NMP does not prevent the development of non-anastomotic biliary strictures in high-risk DCD organs, and our outcomes suggest that extending the donor warm times beyond the currently widely accepted limit of 30 min is not advisable. This finding was not anticipated at the time of trial design or during the conduct of the trial and only became evident during the long-term follow up of these grafts beyond the primary end point of 90 days. Further work is needed to identify new limits (e.g. donor characteristics, warm ischaemia time and cold ischaemia time) and to define perfusion biomarkers that predict this complication and avoid futile transplantation. Recently published research suggests that the composition of bile produced during perfusion (pH, bicarbonate and glucose

**Table 2 Transplant recipient and graft characteristics (median, interquartile range).**

| Recipient characteristics | Trial patients (n = 22) | | | |
|---|---|---|---|---|
| Age in years | 56 (46–65) | | | |
| Sex—n (%) | | | | |
|  Female | 8 (36.4) | | | |
|  Male | 14 (63.6) | | | |
| Body mass index | 28.5 (24.0–31.0) | | | |
| UK end-stage liver disease score | 52 (49–55) | | | |
| Model for end-stage liver disease score[a] | 12 (9–16) | | | |
| Transplant indication—n (%) | | | | |
|  Alcohol-related liver disease | 8 (36.4) | | | |
|  Non-alcohol steatohepatitis | 4 (18.2) | | | |
|  Hepatitis C virus | 2 (9.1) | | | |
|  Primary biliary cirrhosis | 2 (9.1) | | | |
|  Primary sclerosing cholangitis | 6 (27.3) | | | |
|  Hepatocellular carcinoma[b] | 3 (13.6) | | | |
| Need for intra-operative CVVH – n (%) | 1 (4.5) | | | |

| Graft and transplant details | Overall (n = 22) | DBD (n = 12) | DCD (n = 10) | p Value[c] |
|---|---|---|---|---|
| Cold ischaemic time (min) | 452 (316–600) | 507 (408–718) | 416 (354–464) | 0.075 |
| Implantation time (min) | 28 (22–35) | 30 (26–38) | 26 (22–35) | 0.390 |
| Machine perfusion time (min) | 587 (450–705) | 629 (509–700) | 549 (424–780) | 0.598 |
| Total preservation time (min) | 1073 (977–1308) | 1170 (1038–1367) | 1000 (874–1097) | 0.075 |
| Post-reperfusion syndrome | 10 (45.5) | 2 (16.7) | 8 (80.0) | 0.008 |

n number, CVVH continuous veno-venous haemofiltration, DBD donor after brainstem death, DCD donor after circulatory death.
Note: Body mass index is the weight in kilograms divided by the square of the height in metres. Donor warm ischaemic time is defined as the period from the systolic blood pressure decrease below 50 mmHg to commencing the aortic cold flush. Cold ischaemic time is defined as the time between the start of the cold flush during retrieval until the start of machine perfusion. Early allograft dysfunction consists of the presence of one or more of the following variables: (1) bilirubin ≥10 mg/dL on postoperative day 7; (2) INR ≥1.6 on postoperative day 7; (3) aminotransferase level (alanine aminotransferase or aspartate aminotransferase) >2000 IU/mL within the first 7 postoperative days[33].
[a]The liver grafts are allocated in the UK based on the UK end-stage liver disease score; the laboratory values of the model for end-stage liver disease score are included for the comparative information only.
[b]The presence of hepatocellular cancer is recorded as a complication of the underlying liver disease mentioned above, and does not impact on the liver allocation algorithm.
[c]Groups compared by Kruskal–Wallis test to assess differences in continuous variables and Fisher's exact test for categorical variables. Due to the small sample sizes and that the statistical comparison tests were not powered, these results should be interpreted with caution.

**Table 3 Post-transplant outcomes.**

| | Study patients (n = 22) | Control patients (n = 44) | Overall (n = 66) | OR (95% CI), p Value |
|---|---|---|---|---|
| Post-transplant outcomes | | | | |
| Primary graft non-function—n (%) | 0 (0.0) | 1 (2.3) | 1 (1.5) | 1.000[a,b] |
| Early allograft dysfunction—n (%) | 7 (31.8) | 4 (9.1) | 11 (16.7) | 5.62 (1.14–27.79), 0.034[e] |
| Renal replacement therapy—n (%) | 4 (18.2) | 11 (25.0) | 15 (22.7) | 0.68 (0.19–2.38), 0.542[e] |
| Intensive care unit stay (days) | 3.5 (3–4) | 2.0 (1–5) | 3·0 (2.5) | 1.02 (0.95–1.10), 0.566[e] |
| In-hospital stay (days) | 10 (8–17) | 9 (8–11) | 10 (8–13) | 1.00 (0.96–1.05), 0.822[e] |
| Clavien–Dindo complication grade ≥3—n (%) | 7 (31.8) | 17 (38.6) | 24 (36.4) | 0.089[a,b] |
| 90-day graft survival—n (%) | 22 (100) | 41 (93.2) | 63 (95.5) | 0.545[a,b] |
| 90-day patient survival—n (%) | 22 (100) | 44 (100) | 66 (100) | Not applicable |
| 1-year graft survival—n (%)[d] | 19 (86.4) | 38 (86.4) | 57 (86.4) | 1.000 (0.18–5.46), 1.000[e] |
| 1-year patient survival—n (%)[d] | 22 (100) | 42 (95.5) | 64 (97.0) | 0.55[a,b] |
| Biliary complication—n (%)[c] | | | | |
|  Anastomotic biliary stricture[d] | 2 (9.1) | 3 (6.8) | 5 (7.6) | 1.44 (0.19–11.12), 0.725[c,e] |
|  Non-anastomotic biliary stricture[d] | 4 (18.2) | 1 (2.3) | 5 (7.6) | 8.00 (0.89–71.58); 0.063[d,e] |

| | DBD livers (n = 12) | DCD liver (n = 10) | Overall (n = 22) | p Value[b] |
|---|---|---|---|---|
| Study patient biliary strictures | | | | |
|  Anastomotic within 6 months[d]—n (%) | 1 (8.3) | 0 (0.0) | 1 (4.5) | 1.000[a] |
|  Anastomotic within 12 months[f]—n (%) | 1 (8.3) | 1 (10.0) | 2 (9.1) | 1.000[a] |
|  Non-anastomotic within 6 months[d]—n (%) | 1 (8.3)[g] | 2 (20.0) | 3 (13.6) | 0.571[a] |
|  Non-anastomotic within 12 months[f]—n (%) | 1 (8.3)[f] | 3 (30.0) | 4 (18.2) | 0.293[a] |

n number, OR odds ratio, CI confidence interval, DBD donation after brainstem death, DCD donation after circulatory death.
Note: The result needs to be interpreted with caution as the control patients did not receive systematic bile duct imaging; in this group one patient developed non-anastomotic biliary strictures, one died 16 months after the transplantation from biliary sepsis and one is alive with a complex hilar stricture not amenable to any therapeutic intervention.
[a]p Value obtained from Fisher's exact test.
[b]Due to the small sample sizes and that the statistical comparison tests were not powered, these results should be interpreted with caution.
[c]The figures represent strictures manifested with cholestasis and elevated liver enzymes.
[d]Data were assessed at scheduled study visits up to and including the 12-month follow-up visit.
[e]p-Values obtained from conditional logistic regression.
[f]Stricture developed in patient suffering from hepatic artery occlusion requiring revascularisation within 24 h following the transplant.

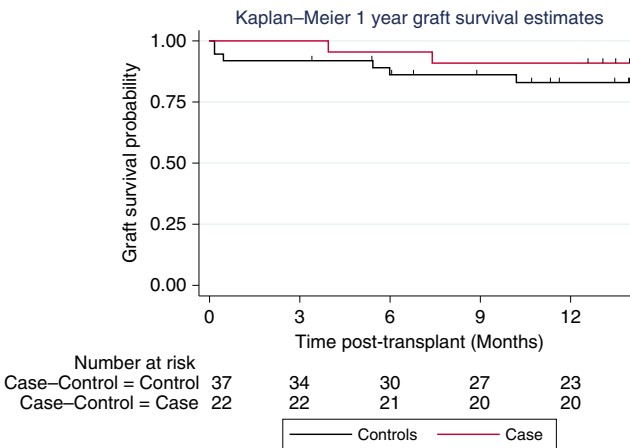

**Fig. 5 Comparison of 1-year graft survival estimate.** Conditional logistic regression was carried out on the matched case–control data to determine the relative risk for graft survival at 1 year between matched case–control groups. The median (range) days follow-up data were included in the survival analyses, but the plot was truncated at 12 months. The ticks on the top of each Kaplan–Meier curve relate to the numbers of patients being censored at that particular time point. There are 2 cases of graft failure in the perfusion group at days 119 and 209; the control group contains 5 graft failures (2 at day 5, 1 at day 14, 1 at day 165 and 1 at day 182). The graft survival was similar in both groups. Findings showed that the odds ratio (relative risk) estimate for graft survival at 6 months was determined as 2.0 (95% CI: 0.2–17.9; $p = 0.535$). Due to the small sample sizes and that this statistical comparison test was not powered, these results should be interpreted with caution.

concentration) is predictive of ischaemic cholangiopathy[17]. Sub-analysis of bile samples and determination of biliary endothelial health is the subject of ongoing research. Evolving novel perfusion strategies might enable the use of DCD grafts exposed to prolonged warm ischaemia[14,29,30].

The other limitations of our study include the sensitivity of the cut-off lactate value, the non-randomised trial design and exclusion of high-risk transplant recipients. Regarding the former, following previous experience, we set the lactate viability threshold to less than 2.5 mmol/L within 2 h of NMP[15,16] To maximise utilisation, this trial extended the assessment period to 4 h. Two livers in the trial were discarded following a rise of the perfusate lactate after meeting the 2-h target. The significance of this is uncertain, although it is notable that a third liver with a similar pattern of lactate clearance was transplanted and experienced a substantial period of early allograft dysfunction with a post-transplant peak ALT of 2074 IU and AST of 3031 IU. Concerning the design, the trial was conducted as a non-randomised study, as transplanting discarded livers with an expected high incidence of primary non-function as controls would be ethically unacceptable. We expect further advances to be achieved through the identification of specific biomarkers that correlate with long-term graft outcomes, in the context of large NMP series or registries. Lastly, as we did not want to compound risks, the study did not include higher risk recipients deemed not suitable to receive marginal organs at the unit's multi-disciplinary liver transplant listing meeting. The majority of participants who decided to participate did so after a long period waiting on the list, with progressive deterioration that was not necessarily reflected by their waiting list position. The feasibility of using livers rescued by NMP for the high-risk recipient is currently under investigation.

In conclusion, this trial demonstrated that NMP provides a way of objectively assessing high-risk organs, and allowed transplantation in a significant proportion of currently unutilised livers without any incidence of primary non-function. The use of perfusion technology was associated with increased graft utilisation, considerably extended preservation time and greatly improved transplant logistics. Adoption of functional assessment of high-risk livers can increase access to life saving transplantation and reduce waiting list mortality.

## Methods

**Study design**. This study was a prospective, open label, phase 2 adaptive single-arm trial comprising high-risk livers meeting two-tier inclusion criteria. The first-tier was being considered as unsuitable for transplant by all UK transplant centres within a nationwide fast-track offering scheme. The trial was performed at a single-institution (Queen Elizabeth Hospital, Birmingham, UK) with experience in NMP and utilisation of high-risk grafts[5,31]. The second-tier eligibility required at least one of seven specific criteria that confirmed the high-risk status of every enroled liver (Table 4). To minimise risks of high post-transplant complications or mortality for the study participants, the trial used an adaptive design with two interim safety analyses (Supplementary Fig. 1). The study design and conduct complied with all relevant regulations regarding the use of human study participants. The trial was funded by the Wellcome Trust, and granted approval by the National Research Ethics Service in London-Dulwich (REC reference 16/LO/1056, Protocol number RG 15–240) and the Medicines and Healthcare Products Regulatory Agency and the University Hospitals Birmingham Research and Development Department. The project was endorsed by the Research, Innovation and Novel Technologies Advisory Group committee of the National Health Service Blood and Transplant. The study was registered at ClinicalTrials.gov (reference number NCT02740608), the protocol has been published[32], and the full version is provided in the Supplementary Information.

**Discarded liver inclusion criteria and the study logistics**. The study considered all potential donors with a diagnosis of brainstem death or Maastricht category III and IV donors after circulatory death, aged up to 85 years, initially retrieved with the intent for transplantation but subsequently declined by all UK transplant centres based on the retrieving or transplant surgeon's assessment. If our centre was the last in the fast-track offering sequence, the liver had to be deemed untransplantable by two consultant surgeons independently. The surgeons were paired together to create an overall low threshold for using marginal livers, ensuring any liver that could be used without viability testing was transplanted, thereby minimising bias. For the liver to be eligible it also had to meet at least one defined high-risk criterion (see Tables 1 and 4). Consent for research was provided by the donor's next of kin.

**Study participants**. Eligible participants were those listed electively for primary liver transplantation and deemed to be low-to-moderate transplant risk candidates, suitable to receive a high-risk graft, as assessed by the unit's transplant waiting list multi-disciplinary team. Candidates were required to have a patent portal vein, no significant comorbidities (cardiovascular diseases, including active angina, a history of ischaemic heart disease, congestive heart failure, cerebrovascular events, symptomatic valvular heart disease or cardiac arrhythmias; pulmonary conditions including pulmonary hypertension or established diagnosis of pulmonary dysfunction), a UK end-stage liver disease[18] (UKELD) score ≤62 and no history of major upper abdominal surgery. Each participant was fully informed of being offered a marginal graft and gave written consent for the trial in advance of the organ offer, after having at least 24 h to consider their participation.

**The study intervention and liver viability assessment**. All livers were cold-preserved with University of Wisconsin solution and commenced NMP using the OrganOx *Metra*™ device after arrival at the transplant centre. The protocol stipulated an NMP duration of between 4 and 24 h. Serial perfusate, bile and tissue samples were taken at regular time intervals. For a liver to be considered viable it had to metabolise perfusate lactate to levels ≤2.5 mmol/L within 4 h of commencing the perfusion, in addition to meeting at least 2 of the following additional criteria: evidence of bile production, maintenance of perfusate pH ≥ 7.30, metabolism of glucose, maintenance of stable arterial and portal flows (≥150 and ≥500 mL/min, respectively), and homogenous perfusion with soft consistency of the parenchyma[16].

If a liver was considered viable, the transplant was set up and performed. At the point of recipient hepatectomy, the NMP team disconnected the organ from the device, flushed it with 3 l of histidine–tryptophan–ketoglutarate solution at 4 °C and handed it over for immediate implantation. Post-transplant management followed the unit's standard protocol, with immunosuppression comprising tacrolimus, azathioprine or mycophenolate mofetil and low-dose steroids. Each patient underwent an MRCP at 6 months unless the investigation was clinically indicated earlier.

Liver quality was determined retrospectively through histological analysis of parenchymal biopsies which were assessed for pre-existing liver disease, steatosis, glycogen content and features of preservation–reperfusion injury.

| Table 4 Study inclusion and criteria. | |
|---|---|
| Graft inclusion criteria | 1. Liver from a donor primary accepted with the intention for a clinical transplantation<br>2. Liver graft was rejected by all the other UK transplant centres via normal or fast-track sequence (see Appendix 3 for list of UK centres)<br>3. One of the following parameters capturing the objectivity of the liver high-risk status:<br>• Donor risk index >2.0[20]<br>• Balanced risk score >9[38]<br>• Graft macrosteatosis >30%<br>• Donor warm ischaemic time (defined as the period between the systolic blood pressure <50 mmHg to the time of commencing donor aortic perfusion) in DCD donors >30 min<br>• Peak donor aspartate and alanine transaminases >1000 IU/mL (AST/ALT)<br>• Anticipated cold ischaemic time >12 h for DBD or 8 h for DCD livers<br>• Suboptimal liver graft perfusion as assessed by a consultant transplant surgeon and documented by graph photography. |
| Graft exclusion criteria | 1. Grafts from patients with active Hepatitis B, C or human immunodeficiency virus infection<br>2. Livers with cirrhotic macroscopic appearance<br>3. Livers with advanced fibrosis<br>4. DCD grafts with donor warm ischaemic time (systolic blood pressure < 50 mmHg to aortic perfusion) more than 60 min<br>5. Excessive cold ischaemic times (DBD > 16 h/DCD > 10 h)<br>6. Paediatric donor (<18 years old)<br>7. Blood group ABO incompatibility |
| Recipient inclusion criteria | 1. Primary adult liver transplant recipient<br>2. Patient listed electively for transplantation<br>3. Low-to-moderate transplant risk candidate suitable for marginal graft, as assessed by the UHB Liver Unit liver transplant listing multi-disciplinary team meeting. |
| Recipient exclusion criteria | 1. High-risk transplant candidates not suitable for a marginal graft<br>2. Patients with complete portal vein thrombosis diagnosed prior to the transplantation<br>3. Liver re-transplantation<br>4. Patients with fulminant hepatic failure<br>5. Blood group ABO incompatibility<br>6. Patient unable to consent<br>7. Patients undergoing transplantation of more than one organ<br>8. Contraindication to undergo magnetic resonance imaging |
| Criteria for transplantation | 1. Lactate ≤ 2.5 mmol/L<br>2. AND two or more of the following within 4 h of starting perfusion<br>• Evidence of bile production<br>• pH ≥ 7.30<br>• Metabolism of glucose<br>• HA flow ≥ 150 mL/min and PV flow ≥ 500 mL/min<br>• Homogenous perfusion |

*DCD* donation after circulatory death, *DBD* donation after brainstem death, *ALT* alanine aminotransferase, *AST* aspartate aminotransferase.
*Note:* Donor risk index is calculated from age, race, cause of death, height and the predicted cold ischaemic time[20]; balanced risk score is calculated using model for end-stage liver disease score (MELD), whether or not the recipient is having a re-transplant or is on intensive care, recipient age, donor age and cold ischaemic time[11].

**Outcome measures**. The co-primary outcomes consisted of (A) feasibility of NMP in discarded organ recovery and (B) achievement of successful transplantation. The perfused organ recovery rate was the proportion of perfusions leading to transplantation. Successful transplantation was defined as 90-day patient survival—a nationally accepted, monitored and continuously audited outcome measure.

The key secondary outcome measures included assessment of the liver graft function (by incidence of primary non-function and early allograft dysfunction[33]) liver function test results, 90-day graft survival, intensive therapy unit and post-transplant in-hospital stays, incidence of vascular complications, and anastomotic and non-anastomotic biliary strictures as assessed by MRCP at 6 months. Perioperative data collection included haemodynamic stability, incidence of post-reperfusion syndrome and blood-product requirements. Post-transplant adverse events and complication severity were graded according to the Clavien–Dindo classification[34]. The secondary outcomes were compared with contemporary controls (1:2), matched in order of priority for the donor graft type, UKELD Score, donor age and donor sex only. Four variables included in the original protocol (model of end-stage liver disease [MELD], recipient age, BMI and the liver disease aetiology) were removed as matching criteria due to confounding correlation and being overly stringent. No recipient suffered from recurrent disease during the study follow-up period. The matching criteria used identified 39 patients. There was consistency in the recipient selection for high-risk grafts guided by the unit's protocols and transplant waiting list multi-disciplinary team meetings that assured similar characteristics regarding the cardiovascular comorbidities and surgical risks in the study participants and the matched controls. The pre-planned comparisons with the matched controls group were not powered to demonstrate any differences. Due to the small sample sizes, these results should be interpreted with caution; the controls were included to present the study results within the context of the unit's contemporary outcomes.

**Statistical analysis**. The trial was powered with an emphasis on (A) the feasibility of the intervention using NMP and (B) recipient safety. In terms of the intervention feasibility (A), the aim was to achieve an organ recovery rate of at least 50%, with a rate of 30% or less being considered unacceptable. Using a two-stage design[35], with an interim assessment after 24 livers (continuing if ≥8 livers were recovered), a sample size of up to 53 livers undergoing NMP might be required, with target alpha (one-sided) of 0.05 (actual alpha = 0.047) and target beta of 0.1 (actual beta = 0.098). NMP was considered feasible for organ recovery if at least 22 livers were recovered from 53 perfused. Though the two statistical inferences are assessing different hypotheses (safety and feasibility), they are linked as 22 transplants are required for the safety testing of the procedure, which is also the minimum number required out of 53 perfused livers to be considered feasible.

For (B), the mean 90-day patient survival rate for patients receiving liver transplants in the United Kingdom was 93%[36]. For the discarded livers, the desirable and undesirable 90-day overall survival rates were set at 88% and 73% (15% lower), respectively. Using an optimal three-stage adaptive design[37] with two interim assessments after 3 patients (requires ≥ 2 successes) and 11 patients (≥8 successes), a sample size of 22 patients was required, with alpha (type I error) and beta (type II error) of 0.2. As this was an early phase (non-definitive) trial to assess the safety of this procedure, a relaxed one-sided alpha was used to attain an achievable sample size within the trial duration and cost constraint. The approach was considered successful if there were at least 18 successes out of 22 transplants.

The descriptive statistics data were presented as number and percentages, and median and interquartile range. Due to small numbers, the pre-planned analyses used Kruskal–Wallis test to assess differences in continuous variables between two groups and Fisher's exact test for categorical variables. Kaplan–Meier survival method was used to analyse time-to-event data and conditional logistic regression for matched case–control analysis. All secondary and exploratory analyses were

two-sided at 5% significance level, not powered and not adjusted for multiple testing. STATA software package version 15.1 for Windows (StataCorp LLC, USA) was used for all analyses. Results were rounded to a relevant precision, percentages in the text to full numbers and p-values to three decimals. The statistical analysis plan is provided in the Supplementary Information.

**Reporting summary**. Further information on research design is available in the Nature Research Reporting Summary linked to this article.

## Data availability

The source data underlying Tables 1–3 and Figs. 4 and 5 are provided in the Source Data File and supplementary tables. Additional data will be provided upon request, according to a procedure described on the Cancer Research UK Clinical Trials Unit website https://www.clinicalstudydatarequest.com.

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

## Acknowledgements

The VITTAL trial represents independent academic research funded by the Wellcome Trust [200121/B/15/Z]. The grant was held by D.F.M., H.M., S.C.A., D.H.A., P.J.F., S.H., J.B. and C.Y. C.Y. was also funded by Cancer Research UK (C22436/A15958). We gratefully acknowledge the support provided by all the team members of the Liver Unit at Queen Elizabeth Hospital Birmingham. In particular we would like to thank the following people and groups: the livers anaesthetic team including John Isaac, Tom Faulkner, Davinia Bennett, Scott Russell, Gowri Subash and Adam Hill, the hepatology team including Philip Newsome, Ahmed Elsharkawy, Andrew Holt, Ye Oo, Neil Rajoriya, Dennis Freshwater, Geoffrey Haydon, Gideon Hirschfield, Fiona Thompson, David Mutimer, Tahir Shah, Shishir Shetty, Dhiraj Tripathi, Matthew Armstrong; the intensive care physician team including Nick Murphy, Tony Whitehouse, Catherine Snelson, Nilesh Parekh, Brian Pouchet and Nandan Gautam; the radiologists Simon Olliff, Brin Mahon, Homoyon Mehrzad and Arvind Pallan. Trial nurses Jo Grayer, Kathryn Rodden, Emma Burke, Francesca Renzicchi and Verity Bratley for their contribution to the study conduct and the data collection. Sara Trevitt, Jennifer Keely, Bhushan Chhajed and the D3B team at the CRUK clinical trials unit at The University of Birmingham. Lorraine Wallace for laboratory support, sample collection and processing. The liver transplant co-ordinators, clinical nurse specialists and allied health professionals at the Liver Unit for help with the study logistics and patient education. Bridget Gunson, Sue Paris and Gary Reynolds for their trial support and work at the NIHR Birmingham Biomedical Research Centre. Leslie Russell, Craig Marshall, Toni Day, Andy Self and Constantin Coussios from OrganOx Ltd for the project partnership and support with the Wellcome Trust funding application. Serena Fasoli and Paul Adams for providing logistical support with OrganOx Ltd materials. David Nasralla and Carlo Ceresa (University of Oxford) for technical support and perfusion advice. NHS Blood and Transplant and the Research, Innovation and Novel Technologies Group (RINTAG) which helped support the implementation of the trial. James Neuberger, Jacques Pirenne, Andrew Hall and Gabriel Oniscu for the trial conduct and safety oversight and membership on the data monitoring committee. Amanda Smith for her help with manuscript preparation and proof reading. In particular we would like to thank also the British Transplant Society, the Liver Advisory Group, the British Liver Trust, and the local public and patient involvement group who supported the trial from its conception, and the Wellcome Trust for funding and supporting this trial. Finally, we would like to thank the organ donors, their families and the 22 study participants, without which this trial could not have been completed successfully.

## Author contributions

H.M., R.W.L., C.Y., S.C.A. and D.F.M. contributed equally. H.M., S.C.A. and D.F.M. conceived the study; C.Y., D.F.M., H.M., A.K. and D.B. designed the trial; R.W.L., M.W.,

A.K., C.Y. and D.F.M. wrote the protocol with contributions from H.M., D.B. and P.J.F.; R.W.L., M.W., S.C. and D.B. were responsible for the trial regulatory documents' preparation and submission; A.K. and C.Y. devised the statistical analysis plan together with D.F.M. M.T.P.R.P.P., P.M., J.R.I., K.J.R., M.A., H.M., A.S., J.F., H.C., J.B., D.H.A., C.M., R.W.L., Y.L.B., J.A. and D.F.M. were involved in the transplantations, machine perfusions and post-transplant patients' management; R.W.L., Y.L.B. and J.A. were responsible for samples and data collection. D.A.H.N. was responsible for histological assessment; A.K. performed the statistical analyses with senior oversight from C.Y.; H.M. wrote the first draft of the paper with input from R.W.L., A.K., C.Y. and D.F.M.; all authors contributed to the study conduct and reviewed the final paper version.

## Competing interests

P.J.F. is a co-founder, chief medical officer and consultant to OrganOx Ltd and also holds shares in the company, he was involved in the study design and funding application, but was not involved in the conduct of the clinical trial. C.M. is an OrganOx Ltd employee and provided the device technical support. H.M. received consultancy fees for lecturing and training activities for OrganOx Ltd. All other co-authors declare no competing interests. This paper presents independent research supported by the NIHR Birmingham Biomedical Research Centre at the University Hospitals Birmingham NHS Foundation Trust and the University of Birmingham. The views expressed are those of the authors and not necessarily those of the NHS, the NIHR or the Department of Health.
