## [Peer Review File · Nature Communications]

Reviewers' Comments:

Reviewer #1:

Remarks to the Author:

The authors have been highly responsive to the reviewers comments and this has significantly improved the manuscript, especially with regard to data interpretation and appropriately framing the significance of the work.

In the response to reviewers, the authors make the statement "We are of the opinion that livers exposed to CIT prior to commencing NMP do not benefit from extended perfusion."

I believe this to be a critical point for the authors to include in the manuscript proper and am certain it will be of interest to those who read this work.

Reviewer #2:

Remarks to the Author:

This is a highly interesting trial, but the significance is lessened because of some limiting factors. The high-risk status of the organs included in the trials is a limiting factor since the authors have used criteria that are not formally established for the definition of marginal organs.

The lack of a true control group is natural considering the concept of the study. However, in reference to the inclusion criteria (e.g. 30% steatosis), it would be reasonable to search the database for liver that fulfilled the high-risk criteria and were transplanted without NMP. The response to the initial review is unsatisfactory. The authors could add a readout according to the established criteria (DRI) and offer a control with a matched DRI group in addition to the selection and data provided.

The approach to employ "a second departmental consultant transplant surgeon's opinion" for the decision making to include an organ into this trial is not adding much value in the objectivity and data driven approach to decision making – unless a formal and well structure process was used. It would be helpful if the authors could clarify the criteria for decision making.

The authors indicate in table S1 which exact criterion for decline was present for each organ eventually not transplanted.

The definition of "low to moderate transplant risk candidates" as indicated in the study participants requires a clearer definition. The control group was not matched for the parameters indicated. A control group matched for a convincing number of parameters should be used if any comparison should be made possible. It is honest to add a comment indicating that the control group is actually not a control group, but to offer the context of the unit's contemporary outcomes is scientifically not very meaningful. The authors refer to the control group as matched controls, but they are not. The authors could try to come up with a better matching control according to the study inclusion and exclusion criteria.

High risk transplant candidate is listed as an exclusion criterion. This needs to be specified. "As assessed by the unit's transplant waiting list multi-disciplinary team" is not sufficiently specific. "No significant cardiovascular comorbidities" should be clarified.

Figure 1B says "Discontinued study following liver re-transplantation at day 225 (n=1) " while the authors conclude (abstract) a 100% (early) graft survival in the abstract. Figure 4 shows 2 graft losses and the abstract (results) indicates 4 graft losses. This is somewhat misleading at first read and could be simplified/clarified. The outcome in the abstract should be changed accordingly. The reasons for graft losses were biliary strictures, requiring retransplantation. It would be helpful to describe the NMP parameters, bile output (during NMP), bile pH (during NMP), course of Bilirubin/INR and Gamma-GT for these four livers to see if there is a correlation. It seems from

table S3 that these livers and/or the postop course were somewhat problematic with longer duration of stay etc.

The fact that the only per protocol MRCP was performed at 6 months possibly leaves some bile

duct injuries hidden. This should be emphasized as a limitation.

Reviewer #3:

Remarks to the Author:

The paper describes a single –arm VITAL trial that aims to assess the viability and success of transplantation of discarded livers. Here are some comments:

Comment 1. Methods, Page 7, line 185. "Acceptance also required...".

Did the authors mean "eligibility criteria" by "acceptance"? Please rephrase the sentence to make it clearer.

Line 187. "To minimise risks...". Please elaborate on the nature of the risks.

Comment 2. Page 9, Line 243. "The secondary outcomes were compared with contemporary controls".

In their reply to comment 5, the authors mentioned that the updated manuscripts dropped these comparisons.

Comment 3. Page 9, Line 264. "actual = 0.047" and "actual = 0.098" should read "actual alpha = 0.047" and "actual beta = 0.098".

Comment 4, Page 9, Line 270. Regarding the Kruskal-Wallis test, why do the authors not use the Mann-Whitney test as there are only two groups?

Comment 5. Page 9, Line 271 "mixed effect modelling". The authors should give more details about the mixed effect model.

Comment 6. Statistical analysis plan, Page 5, "Objectives and Outcome Measures".

The authors use inconsistent language: "1a Establish", "1b Achievement", "Assess", "To compare".

Comment 7. Study Protocol, Page 9, Trial Design.

"Using a Simon's two-stage design (1989)" - a reference number is missing here and in a few other places across the protocol. "with alpha = 0.05 and beta = 0.1." – alpha and beta should be explained.

Comment 8. Study Protocol, Page 39, Statistical considerations".

Inconsistent language and capitalisation: (A) establish... (B) Achievement

"The significance level alpha is set at 0.05, corresponding to the probability of incorrectly rejecting H0 given it is true (Type I error), and the power is set at 0.90 (Type II error rate, beta = 0.10), corresponding to the probability of correctly deciding the NMLP treatment is successful given the true response rate is greater than 50%."

The explanation of alpha and beta should appear earlier in the protocol.

Comment 9. Study Protocol, Page 40. "An optimal three-stage design[17]" A space is missing between the "design" and the reference number.

Comment 10. Study Protocol , Page 41, Secondary Analysis.

"To model repeated measures over time (e.g. quality of life), a linear mixed effects model (taking into account subject correlation) using parametric and more flexible models may be considered."

This sentence was mentioned in comment 16 but has not been addressed.

Comment 11. Inconsistent usage of "beta" and "β" across the protocol.

Comment 12. Inconsistent usage of "H0" on page 39 and "null hypothesis" on page 40. The authors should use consistent notation across the protocol.

Reviewer #1:

The authors have been highly responsive to the reviewers' comments and this has significantly improved the manuscript, especially with regard to data interpretation and appropriately framing the significance of the work. In the response to reviewers, the authors make the statement "We are of the opinion that livers exposed to CIT prior to commencing NMP do not benefit from extended perfusion." I believe this to be a critical point for the authors to include in the manuscript proper and am certain it will be of interest to those who read this work.

We would like to thank the reviewer for the constructive feedback provided for the initial manuscript submission, and for the positive comments regarding the updated submission. We have followed the recommendation, and the latest manuscript version includes a sentence clarifying our experience regarding the perfusion duration required for liver resuscitation.

Reviewer #2:

This is a highly interesting trial, but the significance is lessened because of some limiting factors.

We would like to thank the reviewer for recognising the potential impact of the VITTAL trial, and for the helpful comments regarding sections in our resubmitted manuscript that might be still lacking clarity for the readers. As the main criticism focuses on the definition of the high-risk livers, in the detailed response to Point 1 below we have provided an explanation of the trial concept and challenges we needed to overcome, together with point-by-point answers to the referee's other comments below.

Point 1. The high-risk status of the organs included in the trials is a limiting factor since the authors have used criteria that are not formally established for the definition of marginal organs.

We are grateful to the reviewer for highlighting this important issue regarding the donor liver quality criteria. This aspect was the main challenge in designing the VITTAL trial, as the considerations for liver transplantability are always multi-factorial, including recipient condition, logistical factors, and the surgeon's (or transplanting centre's) experience and their attitude to being risk averse.

The transplant programme at the Liver Unit in Birmingham performs over 200 adult liver transplants annually, and our group has a well-established record and expertise in utilisation of marginal organs. From our experimental and pilot clinical experiences with viability assessment on discarded livers, we learnt that some organs are declined for transplantation for reasons that might not be entirely apparent and widely acceptable, for example due to worsening of the recipient condition, or unfavourable logistics at the transplant centre.

For the VITTAL trial, our team genuinely aimed to push the boundaries of utilisation of truly high-risk organs by accessing the benefit of rigorous peer-review and continual appraisal within the framework of a clinical trial. We included only organs that our team did not feel comfortable to use otherwise, and this attitude was reflected by the two-tier liver inclusion process embedded in the trial design. This was demonstrated by the fact that 25 livers rejected by all centres and offered to us for research were not enrolled to the study for being considered not sufficiently beyond criteria.

We informed the potential participants regarding the high-risk nature of the project and the unknown long-term outcomes of resuscitated livers. Several patients were rather reluctant to participate in the trial, and the lack of sufficient numbers of suitable consented recipients was the principal rate limiting factor in the trial.

Regarding the pre-defined high-risk inclusion criteria, to the best of our knowledge these do not exist and for the purpose of the trial we used an approach demonstrated below with the example of donor transaminase levels.

We initially discussed conservative values, for example transaminases elevated by 3-5x upper limits of normal (ULN; i.e. 150 IU/ml - 250 IU/ml). All the Unit's surgeons agreed that transaminases up to 300-500 IU/ml would be acceptable in donors after brain death and could be transplanted without a high risk of primary non-function. This led us to set a significant upper limit that would be perceived as very high risk by the most critical peers, so set the limit to 1000 IU/ml (approximately 25x ULN, double the limit suggested as potentially acceptable). Some of those organs might still occasionally be used, but this would depend on many other factors including sequential measurement trends, age of the donor, macroscopic liver appearance, microscopic assessment, cold ischaemia and the recipient condition.

We used a similar approach to decide all the other variables, with the exception of the Donor Risk Index and the Balanced Risk Score, both of which are rarely used in the UK; here we sought experience from the literature and overseas colleagues. As can be seen in Table S1, all the calculated risk score (UK-DLI, ET-DRI and UK-DCD) results support the very "marginal" nature of these initially rejected grafts.

Because the principal VITTAL trial inclusion criterion was that every single liver had to be rejected by all the liver transplant units in the UK and discarded for clinical transplantation, we assumed that meeting this condition, in combination with the livers being accepted within a nationwide, non-clinical fast-track allocation scheme used for research organs, would be perceived by peers as a robust enough 1st tier inclusion of poor-quality livers.

Whilst this critical aspect of the VITTAL trial manuscript can be only better clarified, we would genuinely welcome any reviewer's advice as to how to more rigorously define "untransplantable" livers for our future projects. Please see some additional comments related to this issue included in response to other comments.

Point 2. The lack of a true control group is natural considering the concept of the study. However, in reference to the inclusion criteria (e.g. 30% steatosis), it would be reasonable to search the database for liver that fulfilled the high-risk criteria and were transplanted without NMP. The response to the initial review is unsatisfactory. The authors could add a readout according to the established criteria (DRI) and offer a control with a matched DRI group in addition to the selection and data provided.

We thank the reviewer for another thoughtful comment, and we would like to provide additional information to clarify the very high-risk inclusion status in terms of the liver steatosis. We would like again to highlight that the 1st tier criterion was a liver discard by the all other UK centres following a fast-track national allocation. The 2nd tier criteria set for "Steatosis >30%" per se does not necessarily characterise an untransplantable liver; many centres may use these organs for a low MELD / low risk recipient if the liver was from a donor after brain death, and/or the cold ischaemic time was relatively short.

A high degree of steatosis, however, was the principle inclusion criterion in 3 (10%) livers only, and these were reported by the local transplant team as having 50%, 80% and 60% macrovesicular steatosis combined with 11hr:55min, 12hr:00min and 6hr:15min cold ischaemia respectively. These organs were retrieved by other surgical teams, discarded based on the proven histological assessment, and subsequently transported for research to our centre. The probability of using these steatotic livers for transplantation without viability testing in such a context is, in our opinion and experience, rather unlikely. For this reason, we feel that a comparison of the results of steatotic livers from our database might not provide readers with any meaningful comparison and practical information. Instead, the

resubmission includes also the data source file that can be used by readers to compare the liver quality and transplant outcome in the context of their own experience.

Point 3. The approach to employ “a second departmental consultant transplant surgeon’s opinion” for the decision making to include an organ into this trial is not adding much value in the objectivity and data driven approach to decision making – unless a formal and well structure process was used. It would be helpful if the authors could clarify the criteria for decision making.

Please see the above comment regarding the trial design and our team’s intention to provide a rigorous assessment of the very high-risk liver status for the trial eligibility. The attitude of the team was to prevent any unnecessary inclusions that might reduce the trial significance. Regarding the “second opinion” consultant surgeon, the Unit agreed to designate surgeon pairs for the VITTAL study duration to ensure consistency regarding utilisation of marginal livers. The second opinion was required only if the liver was allocated to our Unit within the clinical fast-track allocation scheme, and was also sought for livers that met the inclusion criteria, but had a rather good macroscopic appearance, and implantation could be achieved within 12-14 hours. These details were not specified in the protocol, and for the limited reproducibility were not included in the manuscript text.

Point 4. The authors indicate in table S1 which exact criterion for decline was present for each organ eventually not transplanted.

We thank the reviewer for indicating the lack of clarity regarding the principle study inclusion criteria and presence of additional risks, for example, steatosis >30%. This information is now added to Table S1, highlighting the principle study inclusion criterion in red, and additional risk factors in amber. The information is available for all livers, including organs that did not meet the viability criteria and were eventually not transplanted.

Point 5. The definition of “low to moderate transplant risk candidates” as indicated in the study participants requires a clearer definition.

This is an excellent suggestion; the common characteristics of eligible recipients for the study are now included in the *Methods* section.

Point 6. The control group was not matched for the parameters indicated. A control group matched for a convincing number of parameters should be used if any comparison should be made possible. It is honest to add a comment indicating that the control group is actually not a control group, but to offer the context of the unit’s contemporary outcomes is scientifically not very meaningful. The authors refer to the control group as matched controls, but they are not. The authors could try to come up with a better matching control according to the study inclusion and exclusion criteria.

We appreciate the comment and concern regarding the control group used to compare the VITTAL trial outcomes, and apologise if our previous response lacked clarity. The comparison of the secondary endpoints with the Unit’s contemporary controls was included in the study protocol, and the pre-defined matching parameters were thought to be relevant for the outcomes. In order of priority, matching parameters included the donor type (ie.DCD vs DBD; this was an essential variable to match to achieve similar recipients and overall procedure risk profile and biliary outcomes), UKELD score (reflection of the liver disease severity), donor age (surrogate for concomitant risk factors), sex, and the recipient body mass index (surrogate for the transplant technical difficulty), and underlying liver disease aetiology (impacting on the long-term graft outcomes).

Despite using a database of 384 patients, the matching process was not able to find similarity in all pre-defined variables, so the presented matched controls were based on donor type, UKELD, age and sex only. The discarded variables were MELD (this parameter, however, is largely overlapping with the included UKELD), BMI and aetiology (no patient suffered from recurrent disease during the study follow up period). The matching criteria used identified 39 patients, and whilst it was less than 1:2 match, we still believe it provides the readers with a relevant comparison. Whilst the controls might not be specifically matched for all intended variables, the included key matched criteria ensure similar overall recipient risk stratification. The Unit's approach to match the donor quality with the recipient conditions, co-morbidities and additional risk factors had been consistent (Laing et al, Am J Transplant 2016) over the last decade, meaning any matched recipient of a DCD liver would not have any history of previous major upper abdominal surgery, known portal vein thrombosis or significant cardiovascular or pulmonary comorbidities.

Point 7. High risk transplant candidate is listed as an exclusion criterion. This needs to be specified. "As assessed by the unit's transplant waiting list multi-disciplinary team" is not sufficiently specific. "No significant cardiovascular comorbidities" should be clarified.

This is again an important comment; the common characteristics of recipients not suitable for the study are now included in the *Methods* section.

Point 8. Figure 1B says "Discontinued study following liver re-transplantation at day 225 (n=1) "while the authors conclude (abstract) a 100% (early) graft survival in the abstract. Figure 4 shows 2 graft losses and the abstract (results) indicates 4 graft losses. This is somewhat misleading at first read and could be simplified/clarified. The outcome in the abstract should be changed accordingly. We thank the reviewer for pointing out the inconsistency in reporting the trial design at different time points. The updated manuscript version now clearly specifies the points as "90-day survival" (primary study endpoint), "12-month survival", and the "whole follow-up period" consisting of the median study follow up of 542 days.

Point 9. The reasons for graft losses were biliary strictures, requiring re-transplantation. It would be helpful to describe the NMP parameters, bile output (during NMP), bile pH (during NMP), course of Bilirubin/INR and Gamma-GT for these four livers to see if there is a correlation. It seems from table S3 that these livers and/or the postop course were somewhat problematic with longer duration of stay etc.

This is a very insightful comment. The used viability criteria and the study samples collection protocol did not include qualitative bile assessment and the bile pH measurements were not readily available. The bile production volume measurements, however, were collected and we added those to the Table S3.

Point 10. The fact that the only per protocol MRCP was performed at 6 months possibly leaves some bile duct injuries hidden. This should be emphasized as a limitation.

The per-protocol MRCP at the 6-month time point has been used in several other transplant and machine perfusion studies. Whilst 80% of the clinically relevant biliary strictures are expected to occur within this time frame, the MRCP at this point might reveal asymptomatic irregularities of the bile ducts with varying clinical significance. This manuscript includes patients' follow-up ranging from 390-784 days, including clinical review and liver function tests. This period should provide a long enough time

for clinical manifestation of non-anastomotic biliary strictures. This aspect has been clarified in the updated manuscript text.

Reviewer #3

The paper describes a single – arm VITAL trial that aims to assess the viability and success of transplantation of discarded livers. Here are some comments:

We would like to thank the reviewer for the detailed review of the manuscript and supportive materials included with the submission. The feedback was very helpful and we were able to address all referees' concerns regarding the statistical aspects of the manuscript.

Comment 1. Methods, Page7, line 185. "Acceptance also required..." Did the authors mean "eligibility criteria" by "acceptance"? Please rephrase the sentence to make it clearer.

The relevant sentence in the manuscript *Methods* has been updated accordingly.

Comment 2. Line 187. "To minimise risks...". Please elaborate on the nature of the risks.

The relevant sentence in the manuscript *Methods* now specifies the risk as "risks of unexpectedly high post-transplant complications or mortality".

Comment 3. Page 9, Line 243. "The secondary outcomes were compared with contemporary controls". In their reply to comment 5, the authors mentioned that the updated manuscripts dropped these comparisons.

We apologise if the updated information and manuscript text was misleading. We dropped the mixed effect modelling initially used for the Quality of Life comparisons, however, we kept conditional logistic regression for case-control analysis to presented the study results within the context of the Unit's contemporary controls. The updated manuscript clearly describes that any analyses and comparisons of secondary endpoints were not powered to demonstrate any differences. The updated manuscript adds a notice that "due to the small sample sizes these results should be interpreted with caution".

Comment 3. Page 9, Line 264. "actual = 0.047" and "actual = 0.098" should read "actual alpha = 0.047" and "actual beta = 0.098".

The relevant sentence in the manuscript *Methods* has been updated accordingly.

Comment 4, Page 9, Line 270. Regarding the Kruskal-Wallis test, why do the authors not use the Mann-Whitney test as there are only two groups?

The explanation regarding this statistical test was provided within the response to the referee, but not included in the manuscript text. We apologise for this omission, and the updated manuscript includes this information.

Comment 5. Page 9, Line 271 "mixed effect modelling". The authors should give more details about the mixed effect model.

The mixed effect modelling was dropped from the updated manuscript version; please also see our response to Comment 3.

Comments 6. Statistical analysis plan, Page 5, “Objectives and Outcome Measures”. The authors use inconsistent language: “1a Establish”, “1b Achievement”, “Assess”, “To compare”.

We thank the reviewer for pointing out the inconsistencies in the study *Statistical analysis plan*. We acknowledge that the document contains also several typos and grammatical errors, however, these do not affect the study procedures and analytical methodology. We are of the opinion that the plan should be seen as an historical document and supplementary submission file rather than a file included in the manuscript review process.

Comment 7. Study Protocol, Page 9, Trial Design.

“Using a Simon’s two-stage design (1989)” - a reference number is missing here and in a few other places across the protocol. “with $\alpha = 0.05$ and $\beta = 0.1$.” – α and β should be explained.

We acknowledge this point. The missing reference is: Simon, R. Optimal two-stage designs for phase II clinical trials. *Controlled clinical trials* 10, 1-10 (1989). The comment about the α and β was addressed within the manuscript *Methods* section (see also Comment 3 above). Regarding the *Statistical analysis plan* correction, please see our response to the Comment 6 above.

Comment 8. Study Protocol, Page 39, Statistical considerations”. Inconsistent language and capitalisation: (A) establish... (B) Achievement “The significance level α is set at 0.05, corresponding to the probability of incorrectly rejecting H_0 given it is true (Type I error), and the power is set at 0.90 (Type II error rate, $\beta = 0.10$), corresponding to the probability of correctly deciding the NMLP treatment is successful given the true response rate is greater than 50%.” The explanation of α and β should appear earlier in the protocol.

We acknowledge this point. The clarification of the α and β significance levels was provided within the manuscript *Methods* section. Regarding the *Statistical analysis plan* correction, please see our response to the Comment 6 above.

Comment 9. Study Protocol, Page 40. “An optimal three-stage design[17]” A space is missing between the “design” and the reference number.

We acknowledge this typo and several other inconsistencies and grammatical errors included in the *Study Protocol*, however, these did not affect the study procedures and conduct. The original protocol and all its versions underwent full regulatory review, and were approved by the MHRA, national ethics committees, and local NHS hospital ethics approval (including typographical errors). Due to the regulatory framework, any suggested corrections in the current protocol (version V2.0C – 1st October 2018) would involve the submission of a non-substantial protocol amendment and would require the formal re-submission and review by the above-mentioned regulatory bodies. We feel that changing the protocol at this late stage in the study will not be useful and we are of the opinion that it should be seen as an historical document rather than a file included in the manuscript review process.

We would be able to resubmit the non-substantial protocol amendment for the regulatory authorities’ approval if this was deemed necessary, and was the only concern preventing the manuscript publication.

Comment 10. Study Protocol, Page 41, Secondary Analysis. “To model repeated measures over time (e.g. quality of life), a linear mixed effects model (taking into account subject correlation) using parametric and more flexible models may be considered.” This sentence was mentioned in comment 16 but has not been addressed.

We acknowledge this point and that it was highlighted previously. Please see our comment regarding the process required for the protocol amendments.

Comment 11. Inconsistent usage of “beta” and “ β ” across the protocol.

We acknowledge this point. Please see our comment regarding the process required for the protocol amendments.

Comment 12. Inconsistent usage of “H0” on page 39 and “null hypothesis” on page 40. The authors should use consistent notation across the protocol.

We acknowledge this point. Please see our comment regarding the process required for the protocol amendments.

Reviewers' Comments:

Reviewer #2:

Remarks to the Author:

The authors should be commended for their efforts in the response to the latest comments.

I am concerned, that many of the limitations of the trial remain and that the study will be criticized and questioned for its value. The explanations given by the authors are reassuring but they are not available to the readers in full length. Maybe some of that communication could be offered in a supplemental material.

It is not that I disagree with the study concept and the use of NMP. The study actually describes a currently broadly applied clinical application of NMP. It is the robustness of the conclusions which can be drawn from the trial, that remains fragile.

It remains unclear if the "control group" was matched for the two key selection criteria (PV thrombosis and CV comorbidities?) I would suggest to consider omitting the "control group" since it is not convincing as a true control.

Line 189: „To minimise risks of unexpectedly high....“ Unexpectedly“ seems odd in this context and should be erased.

Reviewer #3:

Remarks to the Author:

I am satisfied with the authors' responses to my comments.

I would like to suggest adding errata for the Statistical analysis plan and the Study protocol, if possible.

Reviewer #2:

The authors should be commended for their efforts in the response to the latest comments. I am concerned, that many of the limitations of the trial remain and that the study will be criticized and questioned for its value. The explanations given by the authors are reassuring but they are not available to the readers in full length. Maybe some of that communication could be offered in a supplemental material. It is not that I disagree with the study concept and the use of NMP. The study actually describes a currently broadly applied clinical application of NMP. It is the robustness of the conclusions which can be drawn from the trial, that remains fragile.

We would like to thank the reviewer for the constructive feedback and for the positive comments regarding the updated submission. We agree that the review process has helped to improve the manuscript clarity and that our correspondence and answers to the reviewers' comments should be made available to the journal readers. We have agreed to publishing these files in our cover letter for this submission.

It remains unclear if the "control group" was matched for the two key selection criteria (PV thrombosis and CV comorbidities?) I would suggest to consider omitting the "control group" since it is not convincing as a true control.

We thank the reviewer for pointing out the lack of clarity regarding the recipients' portal vein thrombosis and cardiovascular comorbidities. The updated manuscript version now includes statement that the matched controls had similar profile with regard to those two specific risk factors. Concerning use of the term "control group", the VITAL study design and protocol used this term, however, we appreciate the reviewer's concern that it might be misleading (for example in a situation reader sees only the manuscript abstract). We have therefore changed the term "control group" to "matched controls" throughout the manuscript text. We believe this is an impartial compromise providing an accurate group description.

Line 189: „To minimise risks of unexpectedly high..." Unexpectedly" seems odd in this context and should be erased.

The relevant sentence in the manuscript *Methods* has been updated accordingly.

Reviewer #3:

I am satisfied with the authors' responses to my comments. I would like to suggest adding errata for the Statistical analysis plan and the Study protocol, if possible.

We are pleased the reviewer has found the updated manuscript acceptable for publication and we would like to thank the constructive feedback provided. The Statistical Analysis Plan and the Study Protocol are included within the manuscript Supplementary Information, and the relevant corrections are included as Errata within their accompanying note.